# High-Dose Fenofibrate Stimulates Multiple Cellular Stress Pathways in the Kidney of Old Rats

**DOI:** 10.3390/ijms25053038

**Published:** 2024-03-06

**Authors:** Agata Wrońska, Jacek Kieżun, Zbigniew Kmieć

**Affiliations:** 1Department of Histology, Faculty of Medicine, Medical University of Gdańsk, 80-210 Gdańsk, Poland; zbigniew.kmiec@gumed.edu.pl; 2Department of Human Histology and Embryology, School of Medicine, University of Warmia and Mazury in Olsztyn, 10-082 Olsztyn, Poland; jacek.kiezun@uwm.edu.pl

**Keywords:** aging, kidney, fenofibrate, cellular stress

## Abstract

We investigated the age-related effects of the lipid-lowering drug fenofibrate on renal stress-associated effectors. Young and old rats were fed standard chow with 0.1% or 0.5% fenofibrate. The kidney cortex tissue structure showed typical aging-related changes. In old rats, 0.1% fenofibrate reduced the thickening of basement membranes, but 0.5% fenofibrate exacerbated interstitial fibrosis. The PCR array for stress and toxicity-related targets showed that 0.1% fenofibrate mildly downregulated, whereas 0.5% upregulated multiple genes. In young rats, 0.1% fenofibrate increased some antioxidant genes’ expression and decreased the immunoreactivity of oxidative stress marker 4-HNE. However, the activation of cellular antioxidant defenses was impaired in old rats. Fenofibrate modulated the expression of factors involved in hypoxia and osmotic stress signaling similarly in both age groups. Inflammatory response genes were variably modulated in the young rats, whereas old animals presented elevated expression of proinflammatory genes and TNFα immunoreactivity after 0.5% fenofibrate. In old rats, 0.1% fenofibrate more prominently than in young animals induced phospho-AMPK and PGC1α levels, and upregulated fatty acid oxidation genes. Our results show divergent effects of fenofibrate in young and old rat kidneys. The activation of multiple stress-associated effectors by high-dose fenofibrate in the aged kidney warrants caution when applying fenofibrate therapy to the elderly.

## 1. Introduction

The aging process is accompanied by molecular alterations that significantly impact renal structure and function, contributing to an increased vulnerability to age-related renal pathologies. Stress-associated molecular pathways, encompassing oxidative stress, osmotic stress, dysregulated signaling of hypoxia, as well as inflammation, stand out as critical players in the age-related decline of renal homeostasis. Within this context, fenofibrate (FF), a peroxisome proliferator-activated receptor alpha (PPARα) agonist used as a hypolipemic drug, has emerged as a potential modulator of these pathways.

FF has been demonstrated to exert positive effects on kidney function in certain clinical contexts. As dyslipidemia is a common comorbidity in individuals with chronic kidney disease (CKD), FF-induced improvements in lipid metabolism (i.e., reductions in serum triglyceride levels and low-density lipoprotein cholesterol, LDL) may contribute to overall renal health, reflected by a reduction in albuminuria and glomerular filtration rate (GFR) loss in type 2 diabetic (T2D) patients [1]. Indeed, FF has been reported to reduce cardiovascular death and delay the need for permanent dialysis in patients with advanced CKD [2], which was recently confirmed in a case–control database study [3]. Likewise, in db/db diabetic mice FF ameliorated renal damage by reducing lipotoxicity [4]. Also, in diabetic rats, FF exerted renoprotective effects and improved endothelial function [5]. FF’s ability to enhance antioxidant defense mechanisms could contribute to its impact on oxidative stress mitigation [5]. Furthermore, FF may help alleviate renal inflammation, a common denominator in various kidney disorders. For instance, FF suppressed inflammation and alleviated renal injury in Zucker diabetic fatty ZD rats [6] and attenuated inflammatory responses in renal cells [7]. Lastly, some studies showed that FF may mitigate renal fibrosis in hypertensive rats [8], as well as in diabetic nephropathy, through the attenuation of transforming growth factor-beta (TGF-β)-induced fibrotic responses [6].

The administration of FF, while generally considered safe and effective for managing dyslipidemia, has been associated with certain adverse effects on kidney function. The elevation of plasma creatinine levels, a simple marker of kidney function, is often observed during FF therapy [9]. The ACCORDION study showed that in T2D patients, long-term intensive FF treatment may increase the risk for adverse kidney events [10].

Older individuals often present with age-related physiological changes and an increased susceptibility to adverse drug reactions. However, there is a scarcity of data concerning the molecular effects of FF treatment on the aging kidney. We therefore undertook this study to investigate the effects of low-dose and high-dose FF on stress-associated molecular pathways in the kidney of young and old rats under standard breeding conditions. In particular, we wanted to examine whether aging affects the FF-induced modulation of antioxidant defenses, hypoxia and osmotic stress responses, molecular effectors of inflammation, as well as metabolic regulators and enzymes.

## 2. Results

### 2.1. Fenofibrate Treatment Increased the Collagenous Tissue Area in Old Rats and Mildly Affected Kidney Function Parameters in Both Age Groups

The kidney tissue, collected from young and old rats from control groups and after 30 days of FF treatment (Figure 1), was analyzed using various histological staining methods. The kidney morphology of control old rats revealed typical aging-related changes of the glomeruli and tubules (Appendix A). Generally, the renal corpuscles in the kidney cortex of old rats were larger than those of the young ones; however, many presented glomerulosclerosis and increased urinary space (Figure 2a, Appendix A). Mallory staining revealed an increase in collagenous compartment both in the glomerular mesangium and interstitium of the aged kidney (Figure 2a). Additionally, in old kidneys some tubuli, presumably proximal convoluted ones, were clearly dilated with a reduced density and height of the brush border (Appendix A). PAS staining revealed an age-related increase in intraglomerular-stained material, thickening of the glomerular basement membrane (GBM), as well as those of Bowman’s capsule and basal laminae underlying the renal tubules (Figure 2b).

Generally, there was little impact of fenofibrate administration on kidney morphology assessed using hematoxylin and eosin (HE) staining in either age group (Appendix A). The 0.1% FF decreased the collagenous tissue area fraction by 46%, assessed by Mallory staining, in the renal cortex of young rats, while 0.5% FF increased the collagenous area in old rats by 62% (Figure 2c). Moreover, in young rats FF treatment did not affect the PAS-stained area fraction of the kidney cortex. In contrast, in old rats fed 0.1% FF (but not the higher dose), the drug noticeably ameliorated the age-related thickening of GBM and other basement membranes (Figure 2d).

Serum creatinine and blood urea nitrogen (BUN) concentrations are widely used to estimate kidney function. Old control rats presented similar levels of serum creatinine and BUN compared to the young ones (Figure 2e). 30-day treatment of young and old rats with the higher FF dose increased serum creatinine level in both age groups, however, the effect was more pronounced in old animals. BUN was not affected by the FF treatment in young rats, whereas in old ones only the lower dose increased the BUN level by 29% (Figure 2e). Thus, FF had a mildly negative impact on parameters of kidney function, more pronounced in the old than the young rats.

### 2.2. Low and High Doses of Fenofibrate Differently Affected the Expression of Stress-Associated Genes

To test whether FF modulates the mRNA expression of targets responsive to cellular stress insults, a dedicated PCR array for 84 genes was used. Analysis of differentially expressed genes (DEGs) revealed striking differences between 0.1% and 0.5% FF treatments (Figure 3). In both age groups, the lower dose mildly downregulated the expression of genes predominantly involved in hypoxia signaling, inflammatory responses, osmotic stress, DNA damage response, oxidative stress, and unfolded protein response (Appendix A). In contrast, the higher FF dose mostly upregulated the tested genes, with a higher number of DEGs in old rats than in young rats. We hypothesize that the dose of FF may therefore determine the molecular response of rat kidney to various stress insults.

### 2.3. Fenofibrate Activated Components of Cellular Antioxidant Defences in the Kidney of Young but Not Old Animals

Because the PCR array revealed antioxidant-associated DEGs responsive to FF treatment, selected components of the antioxidant response were analyzed using real-time PCR and Western blotting techniques (Figure 4). First, we determined the expression of Nuclear factor erythroid 2-Related Factor 2 (NRF2), a transcription factor that activates the expression of antioxidant response genes [11]. The mRNA (Figure 4a) and protein levels (Figure 4b) of NRF2 were similar in the renal cortex tissue of young and old rats. Both doses of FF decreased mRNA levels of *Nrf2* in young but not old animals (Figure 4a), however, FF did not affect NRF2 protein levels in either young or old rats (Figure 4b). Next, we analyzed the protein levels of sestrin 2 (SESN2), which stimulates NRF2 by targeting its repressor Kelch-like ECH-Associated Protein 1 (Keap1) for autophagosomal degradation [11,12]. *Sesn2* mRNA expression levels were similar in young and old control rats (Figure 4a), whereas SESN2 protein levels were two-fold higher in the old animals (Figure 4b). In both young and old animals, 0.5% FF decreased *Sesn2* expression in the kidney (Figure 4a). In young rats, both doses of FF increased SESN2 protein level to a similar extent, however, in the kidneys of old animals the higher dose of FF tended to decrease SESN2 expression (*p* = 0.12) (Figure 4b).

In addition, we measured the mRNA levels of NRF2 target genes, Heme Oxygenase 1 (*Ho-1*) and Sulfiredoxin 1 (*Srxn1*). The expression of both *Ho-1* and *Srxn1* was induced by low-dose FF in young rats (Figure 4a). However, in old rats either dose of FF failed to induce any changes in *Ho-1* and *Srxn1* expression at the mRNA level (Figure 4a).

To assess the impact of FF on oxidative stress at tissue level, we measured the immunoreactivity of 4-hydroxynonenal (4-HNE), which is the end product of membrane lipids’ peroxidation [13] and a marker of oxidative stress. 4-HNE immunoreactivity was detected throughout distal convoluted tubules and less intensely in the proximal tubules, where it localized primarily near the apical surface of the epithelium (Figure 4c). Old control rats presented lower 4-HNE immunoreactivity than the young ones. In the kidney cortex of young rats, high-dose FF significantly reduced 4-HNE levels. On the contrary, in old rats, 0.5% FF increased 4-HNE levels (Figure 4c). We conclude that FF-associated activation of cellular antioxidant defenses may be impaired in the kidney of old rats.

### 2.4. Fenofibrate Modulated the Expression of Factors Involved in Hypoxia and Osmotic Stress Signalling in a Similar Manner in Young and Old Rats

Verification of the results of the PCR array revealed some interesting observations regarding the expression levels of the genes related to hypoxia and osmotic stress. Vascular Endothelial Growth Factor A (VEGFA), produced in the kidney predominantly by podocytes, plays vital roles for glomerular endothelial cells’ and podocytes’ physiological functions [14]. The mRNA levels of *Vegfa* in the rat kidney were altered by both doses of FF in a similar manner: the levels in control animals did not differ between age groups, whereas 0.1% FF decreased, and 0.5% FF returned *Vegfa* expression to the control values (Figure 5a). This pattern of response was challenged at the protein level since the lower FF dose increased VEGF expression in both age groups, whereas the higher FF dose increased VEGF in the kidney of young rats and decreased in old animals. Furthermore, in both age groups of animals low-dose FF reduced the expression of *Serpine1* (encoding Plasminogen Activator Inhibitor 1, PAI-1) (Figure 5a), which is under transcriptional control of Hypoxia Inducible Factor 1 (HIF-1) [15]. *Serpine1* expression was also inhibited by high-dose FF in old rats (Figure 5a).

FF modulated the expression of osmotic stress-associated genes (Figure 5b). The mRNA level of Cystic Fibrosis Transmembrane Conductance Regulator (*Cftr*) gene was two-fold lower in old than in young control rats, whereas the expression levels of Solute Carrier Family 9 Member A3 (*Slc9A3*) and Endothelin 1 (*Edn1*) genes were similar in both age groups (Figure 5b). However, FF affected the kidney expression of these genes in dependance on the animals’ age. In young rats, both doses of FF reduced the mRNA expression of *Cftr*, and only 0.1% FF reduced the gene’s expression in old rats. On the contrary, in young and old animals only 0.1% FF increased *Slc9A3* mRNA levels, while only 0.5% FF increased *Edn1* expression in both age groups (Figure 5b).

### 2.5. Inflammatory Response Genes Were Differently Affected by Fenofibrate in Young and Old Rats

Since the PCR array revealed variable modulation of inflammatory genes’ expression (Appendix A), we evaluated the content of a key inflammasome component [16], NLR family Pyrin domain containing 3 (NLRP3) protein, and its gene expression. The levels of NLRP3 mRNA and protein did not differ between young and old control rats (Figure 6a). Treatment with 0.1% FF (but not 0.5% FF) tended to increase *Nlrp3* mRNA expression in the kidney of young animals (*p* = 0.059), and significantly increased the expression in the old ones (Figure 6a). However, NLRP3 protein content in the kidney did not change significantly in either young or old rats treated with FF (Figure 6a). We also measured the mRNA and protein expression of the proinflammatory cytokine and marker of inflammaging IL-6 [17]. *Il-6* mRNA expression was higher in the kidney of old than in young control rats, however, there were no significant differences in the protein levels (Figure 6a). Treatment with either dose of FF increased *Il-6* expression in the animals, with a higher increase after the low dose in young rats, and after the high dose in old rats. However, kidney IL-6 protein levels were not affected by FF treatment in young and old animals (Figure 6a).

We verified by qPCR the expression of some other DEGs involved in the inflammatory responses (Figure 6b). The basic mRNA expression levels of Caspase 1 (*Casp1*), Interleukin 1b (*Il-1b*) and Chemokine (C-C motif) ligand 12 (*Ccl12*) were similar in the kidney cortex of young and old rats. In the kidney of young rats, 0.5% FF (but not its lower dose) decreased the expression of *Casp1*, both doses of FF decreased *Il-1b* gene expression, but neither dose affected *Ccl12* expression (Figure 6b). In the kidney cortex of old rats, *Casp1* expression was likewise reduced after 0.5% FF. In contrast to young rats, the expression of the proinflammatory gene *Il-1b* was not significantly affected, and that of *Ccl12* increased after 0.5% FF treatment (Figure 6b).

We next performed immunohistochemical staining for Tumor Necrosis Factor-alpha (TNFα), a major proinflammatory cytokine expressed primarily by macrophages, monocytes, neutrophils, T cells, and NK-cells, but also at low level by epithelial cells of kidney tubules [18,19]. Indeed, diffuse TNFα immunoreactivity was present in some segments of renal tubular cells (Figure 6c, Appendix A). Similar levels of immunoreactivity were observed in young and old control animals. Treatment with FF had little impact on TNFα immunoreactivity in young rats. However, old animals administered 0.5% FF presented significantly elevated TNFα immunoreactivity, including the apical surface of proximal convoluted tubules, which may represent the receptor-bound antigen molecules (Figure 6c).

### 2.6. Effects of Fenofibrate on Kidney Nutrient-Sensing and Metabolic Effectors

The basic molecular mechanisms of FF action results from its binding to and activation of PPARα, leading to the activation of the key cellular energy sensor AMP-activated protein kinase (AMPK), as well as its target Peroxisome proliferator-activated receptor Gamma Coactivator 1 alpha (PGC1α), which is a master regulator of mitochondrial energy metabolism [20,21]. We compared how two doses of FF affected the expression of regulators and enzymes for fatty acid oxidation (Figure 7). In the kidney of young rats treatment with 0.1% FF tended to increase (*p* = 0.1) the nuclear abundance of the active phospho-AMPK (Thr172), as well as the nuclear levels of PGC1α, which undergoes phosphorylation and thus activation by AMPK [22]. Both these effects of 0.1% FF were significant in old rats (Figure 7a). However, treatment with 0.5% FF did not significantly affect pAMPK or PGC1α levels in the rat kidney. AMPK may also indirectly activate sirtuin 1 (SIRT1), a histone deacylase involved in many regulatory cellular processes and implicated in the control of cellular aging [23]. The level of SIRT1 was upregulated by 0.5% FF in the kidney of young but not old rats (Figure 7a), even though *Sirt1* mRNA expression increased after treatment only in the old animals (Figure 7b).

The protein and mRNA levels of the mitochondrial β-oxidation enzyme medium chain acyl-CoA dehydrogenase (MCAD) were lower in old than in young rats under basic conditions. Both doses of FF elevated MCAD protein levels in both age groups of rats (Figure 7a). However, FF treatment in young animals had little effect on the mRNA expression of *Mcad*, as well as other genes involved in fatty acid oxidation (long-chain acyl-CoA dehydrogenase, *Lcad*; carnitine palmitoyltransferase 1, *Cpt1*; peroxisomal acyl-coenzyme A oxidase 1, *Acox1*). In contrast, in old rats FF mildly increased the enzymes’ mRNA levels (except for *Lcad*) (Figure 7b). In both age groups, the expression of *Hmgcs2* gene (hydroxymethylglutaryl-CoA synthase 2), encoding a key enzyme of ketogenesis, was highly induced by the treatment with 0.1% FF, however, this effect was suppressed in rats treated with 0.5% FF (Figure 7b). Our data show that in the young and old rat kidney FF upregulates the fatty acid oxidation enzyme MCAD at the protein level, but only in old rats increases the involved genes’ transcript levels.

## 3. Discussion

Fenofibrate, a well-established lipid-lowering drug, has garnered recognition for its pleiotropic benefits in various pathological contexts, including diabetes. Because metabolic derangements become more common with advancing age, our investigation aimed to compare the renal effects of two different fenofibrate doses in young and old rats. We focused on FF’s impact on the kidney structure and expression of molecular effectors involved in cellular stress responses. Our results indicate that high-dose fenofibrate may negatively affect the kidney morphology and activate multiple stress pathways in the aged kidney. Conversely, low-dose fenofibrate treatment in old rats activates fatty acid oxidation enzymes’ expression more significantly than in the young ones. These age-related distinctions in renal responses to fenofibrate emphasize the need for precise therapeutic dosage determination, underscoring the importance of nuanced administration in aging populations.

We chose to compare young-adult and old male rats on a balanced diet supplemented with FF for 30 days, which causes the hypolipemic effect, as we previously described [24]. Here, we present novel data concerning the aging-associated distinctions of FF effects in the kidney. Even though some renoprotective effects of FF have been demonstrated in rodent models of various pathologies [8,25,26,27], to our best knowledge, there have been no studies comparing the impact of FF in the kidney of young and old animals. We found only one study that investigated the renal FF effects in old mice, in which a longer therapy (0.1% FF in chow for 6 months) ameliorated age-related renal injury [28].

Our histological analyses confirmed the previously described aging-associated changes in the human and rodents’ kidney structure, including glomerulosclerosis, interstitial fibrosis, expansion of mesangium, and thickening of basement membranes [29,30]. The 30-day-long supplementation of old rats with 0.1% FF ameliorated the thickening of basement membranes and mesangial expansion, similarly to amelioration of high fat diet (HFD)-induced kidney changes in mice treated with FF (50 mg/kg b.w.) [25]. However, we did not find such effects after 0.5% FF dose, nor in the young animals. Moreover, we observed an anti-fibrotic action of the lower, but not the higher FF dose only in young rats. This result aligns with reports of FF’s renal anti-fibrotic effects in rodent models of HFD [25], diabetes [6,27,31], ischemia-reperfusion injury [32], and renal transplants [33,34]. Indeed, it has been suggested that renal lipotoxicity due to elevated blood lipid levels causes transcriptional suppression of PPARα by ATF6α (Activating Transcription Factor 6 alpha), leading to increased expression of connective tissue growth factor involved in fibrosis [32]. Strikingly, in the old rats in our study, not only did 0,1% FF fail to reduce collagenous tissue area, the 0.5% dose even expanded the fibrotic tissue. In contrast to our findings after 30-day treatment, a 6 months intervention with 0.1% FF caused significantly less tubulointerstitial fibrosis and less expansion of the mesangial area as compared to control old mice [28]. Because tubulointerstitial fibrosis strongly contributes to chronic kidney disease, our results imply that caution and appropriate dosage are essential for FF therapy in the elderly.

We compared how the two FF doses affected the expression of stress-associated genes using a dedicated PCR array. In both age groups of rats, the opposite direction of changes induced by 0.1% and 0.5% FF indicates that the drug dosage may differently affect the molecular stress responses in the kidney. Indeed, the subsequent molecular analyses partially confirmed that high-dose FF, in contrast to the low dose, caused exacerbation of oxidative stress and inflammatory genes’ expression in the old rats’ kidney, as discussed below.

Chronic kidney failure, which often occurs in the elderly, may be accompanied by diabetes, in particular T2D [20]. Oxidative stress is a pivotal factor in the pathogenesis of diabetes [35]. The antioxidant protective mechanisms of FF have been demonstrated in rodent models of kidney damage associated with streptozotocin-induced type 1 diabetes [36,37,38] and high fat diet [25]. For example, in the kidney of 8-week-old mice, FF increased the nuclear accumulation of NRF2 as well as the expressions of its downstream antioxidant genes including *Ho-1* and *Nqo1* (NAD(P)H Quinone dehydrogenase 1), both in diabetic and non-diabetic animals [31]. Our study of FF’s effects in the kidney of young and old rats confirms and expands on these findings. We demonstrated limited antioxidant effects of FF in young, normally bred rats, in which low-dose FF induced the antioxidant genes *Ho-1* and *Srxn1*, but not the protein levels of NRF2. Interestingly, only in young rats, we found high-dose FF to suppress the immunoreactivity of 4-HNE, a marker of oxidative stress. In comparison to our results, the data from non-diabetic FF-treated mice (100 mg/kg b.w. every other day for 3 or 6 months) in the study by Cheng et al. (2016) [31] does not show any change in 4-HNE immunoblot levels. Of note, the authors demonstrated a critical role of Fibroblast Growth Factor 21 (FGF21) in the protection conveyed by FF against diabetic nephropathy, as well as the activation of NRF2 through Akt2 signaling [31,36]. Our study provides novel data obtained in old rats, showing that high-dose FF failed to induce antioxidant protection in the old kidney, exacerbating oxidative stress. This suggests an impairment in FF’s activation of cellular antioxidant defenses during kidney aging.

In our model, FF age-independently elevated the renal levels of VEGFA protein, despite decreased *Vegfa* mRNA after low-dose FF, suggestive of post-transcriptional mechanisms of regulation. Using bioinformatics tools, FF has been previously predicted to increase VEGFA expression in diabetic nephropathy [39], while the drug has been shown to increase VEGFA levels in the visceral adipose tissue of FF-treated normolipemic rabbits [40]. Moreover, in both young and old rats in our model, low-dose FF inhibited the expression of *Serpine1*, activated by HIF-1 under hypoxic conditions [15]. Such an inhibitory effect by FF has been previously reported in the renal cortex of diabetic rats [27] and in human umbilical endothelial cells [41]. The analysis of aging effects on the transcriptome of renal endothelial cells (ECs) using single-cell RNA sequencing revealed increased gene expression (and immunohistochemical staining) of SERPINE1/PAI-1 in the glomerular ECs of old mice [42]. Probably because of lower sensitivity of our microarray study of kidney cortex homogenates, we could not confirm such age-associated changes at the mRNA level. Further studies of kidney function are needed to test if FF may favorably affect renal vascularization and perfusion.

In both age groups FF exhibited consistent effects on osmotic stress-related factors, including *Cftr*, *Slc9A3*, and *Edn1* expressions. The decrease in *Cftr* expression by FF is likely due to the activation of AMPK, since AMPK activation by metformin has been shown to inhibit *Cftr* in models of autosomal dominant polycystic kidney disease [43]. The plasma membrane Na(+)/H(+) antiporter *Slc9A3*, which plays a key role in salt and fluid absorption and pH homeostasis, was upregulated by the low-dose FF. This is the first report of FF effect on *Slc9A3* expression, besides the study on intestinal epithelium in mice fed a low-fat diet and FF for 12 days, which reported no change in SLC9A3 (NHE3) protein level [44]. In comparison to *Slc9A3* modulation, it was the high dose of FF that increased the expression of endothelin 1 gene in the kidney of both young and old rats. In contrast, in cultured human microvascular endothelial cells FF suppressed *Edn1* mRNA and protein expression, via PPARα-dependent and independent mechanisms [45]. This inhibitory action of FF was suggested as a therapeutic modality against pulmonary hypertension in sickle cell disease [46] and against endothelin-induced cardiac hypertrophy [47]. The discrepancies between the results of our in vivo study and in vitro obtained data merit further studies to assess the (patho)physiological context of various experimental models.

Our data on modulation of inflammatory factors by FF underscores the contrasting and dose-associated outcomes in the young and old rats. Young animals treated with FF exhibited a dampened proinflammatory response, indicated by the reduced gene expression of the inflammasome-activated caspase *Casp1* and the cytokine *Il-1b*. However, other components such as NLRP3 and IL-6 protein levels, as well as TNFα immunoreactivity, remained unaffected by FF in the young rat kidney.

Our study unveils that in the aged kidney high-dose FF may contribute to the promotion of the inflammatory response, manifested as elevated TNFα immunoreactivity and increased expression of some proinflammatory genes. The formation of TNFα, which regulates hemodynamic and excretory functions in the kidney, increases oxidative stress and may promote renal dysfunction [18]. However, despite signs of oxidative stress in the high-dose FF-treated old rats, there were no changes in the expression of NLRP3, a crucial component of the inflammasome which can be activated by danger-associated molecular patterns (DAMPs) during oxidative stress [48]. Our findings underscore a potential risk of FF therapy exacerbating inflammaging in the elderly [49,50], in particular, worsening conditions like chronic kidney disease [23].

In contrast to our results, previous studies have reported consistent anti-inflammatory effects of FF in pathological settings in laboratory rodents. Examples include anti-inflammatory action of FF in HFD-induced kidney inflammation [25], hypertension in rat [8], renal ischemia-reperfusion injury in mouse [51], doxorubicin- or cisplatin-induced nephrotoxicity in rat [37,38,52]. The anti-inflammatory effects of FF have also been observed in obese patients with T2D [53]. On the other hand, data from control FF-treated rats in many studies showed no such anti-inflammatory effects under non-pathological conditions [37,38]. These divergent outcomes may be explained by the anti-inflammatory effect of FF being secondary to its hypolipemic effect, although direct regulation of inflammatory signaling pathways plays a role in some settings [54].

FF activation of PPARα is known to stimulate oxidative metabolism, which was confirmed in our study, in line with several other studies in rodents [4,5,25]. Notably, the lower FF dose was more effective than the higher dose in activating AMPK and PGC1α. Interestingly, SIRT1 protein, which can be activated by AMPK, was elevated only in young rats treated with 0.5% FF. The old animals, however, presented increased *Sirt1* mRNA expression, suggestive of age-related alterations in post-transcriptional regulation. For example, Chalkiadaki et al. [55] demonstrated SIRT1 cleavage by CASP1 in adipose tissue of mice fed high-fat diet. However, in our study we did not measure CASP1 protein levels, whereas its gene expression was in fact reduced after 0.5% FF treatment. Additionally, in our study FF demonstrated a more significant influence on β-oxidation enzymes’ gene expression (*Mcad*, *Acox1*) in aged rats compared to their younger counterparts. In contrast, aging did not affect the FF induction of the ketogenic gene *Hmgcs2*, which is known to be under PPARα transcriptional regulation [56]. Other studies also showed induction by FF of the protein or mRNA levels of *Acox1*, *Mcad*, and *Cpt1* in young mice fed HFD [25] or in a mouse model of autosomal dominant polycystic kidney disease [57]. We suggest that the age-related differences in the metabolic effectors’ modulation by FF may arise from the higher lipid levels in old than in young rats, as shown in our earlier study [24].

The present study had several limitations which may affect interpretation of our results in a broader context. First of all, further studies are needed to identify the aging-associated molecular mechanisms responsible for the diverse FF outcomes in the kidney of young and old rats. Moreover, important kidney function parameters, including clearance of creatinine and glomerular filtration rate, as well as activity assays for oxidative metabolism enzymes, and measurements of malondialdehyde and reduced glutathione levels are needed to validate our findings. Furthermore, PCR array indications that high-dose FF affects genes involved in apoptosis, necrosis, genomic stress, and DNA repair (Appendix A) give ground for further research, aiming to explain some adverse effects of 0.5% FF in the aged kidney.

## 4. Materials and Methods

### 4.1. Animal Experiments

The experiments were conducted on male Wistar-Han rats bred under standard conditions in the Academic Animal Experimental Centre in Gdansk, Poland. All animal experimental procedures had been authorized by the Local Ethical Committee in Bydgoszcz (protocols No. 41/2017, 58/2017, 40/2018, and 5/2019) and were conducted in accordance with the institutional and European guidelines for animal experimentation.

Young adult (4-month) and old (24-month) male rats were fed either standard rodent chow (Labofeed H, Wytwornia Pasz Morawski, Kcynia, Poland; control animals, *n* = 8–10 animals in each young and old group) or for 30 days received FF mixed into the chow before pelleting (Glentham Life Sciences, Corsham, UK; FF-treated groups, *n* = 8–10 animals per group). The chow contained 22% crude protein, 3.5% crude fiber, 4.2% crude fat, and 37% starch. Two doses of FF were tested in separate experiments. First, the dose of 0.5% (*w*/*w*) FF was introduced to provide approximately 100 mg/day FF according to the baseline food intake (20.725 ± 2.43 g and 23.39 ± 4.97 g, respectively in young and old rats). Because of the mild hepatotoxicity of the 0.5 FF dose [24], the effects of a lower FF dose (0.1% FF, *w*/*w*) were subsequently tested. The 0.5% FF dose was equivalent to 260 and 210 mg/kg body weight in young and old rats, respectively, and the 0.1% FF dose—52 and 42 mg/kg. Food intake and, in consequence, FF dose, were monitored every second day throughout the experiment by measuring the weight of chow left from pre-weighted samples (ad libitum feeding). After 30 days of the treatment, overnight-fasted animals were sacrificed under full anesthesia (5% isoflurane inhalation) through exsanguination from heart puncture. The kidneys were collected and cut into smaller samples, immediately frozen in liquid nitrogen and stored at −80 °C or fixed in buffered 4% paraformaldehyde.

### 4.2. Serum Concentrations of Creatinine and Blood Urea Nitrogen

Measurements of creatinine and blood urea nitrogen (BUN) levels in fresh blood sera were performed in routine diagnostic tests by the Veterinary Diagnostic Laboratory Lab-Wet (LabWet, Gdańsk, Poland).

### 4.3. Histological Stains

Formaldehyde-fixed kidney samples were embedded in paraffin using a tissue processor (STP 120, Zeiss, Jena, Germany). Four-μm-thick sections, obtained using a rotary microtome (Opta-Tech MR-315, Warsaw, Poland), were mounted on glass slides (SuperFrost Ultra Plus, Thermo Scientific, Vantaa, Finland) and stained with hematoxylin and eosin (HE), periodic acid-Schiff (PAS), and Mallory trichrome methods, following standard protocols [58]. The HE and PAS stains were used to assess age-related and treatment-induced changes in the kidney’s morphology and presence of pathologies. Collagenous tissue area fraction was assessed using the Mallory stain.

The slides were inspected using a microscope (Olympus Cell-Vivo IX 83, Olympus Corp., Tokyo, Japan) equipped with a digital camera (SC-50, Olympus Corp.), and microphotographs were obtained at 10×, 20× and 40× objective magnifications. Measurements of Mallory-stained collagenous tissue area fraction were performed for four rats in each group, in at least six non-overlapping fields on non-contiguous sections, using a neuronal network of the cellSens Dimension 4.1 software (Olympus Corp.). Measurements of the PAS-stained area were performed using the Count and Measure function of the cellSens Dimension 4.1 software (Olympus Corp.) for RGB-separated images with appropriate threshold set up, for 4-7 rats per group, 6 non-overlapping fields for each.

### 4.4. Immunohistochemical Staining for Tumor Necrosis Factorα (TNFα) and 4-Hydroxynonenal (4-HNE)

Immunohistochemical (IHC) analysis was performed as described previously by Kieżun et al. (2022) with modifications [59]. The sections were subjected to an antigen retrieval procedure by microwaving for 6 min in Retrieval Solution Buffer, pH 6.0 or pH 9.0 (Leica Microsystems, Wetzlar, Germany; pH 6.0 for TNFα and pH 9.0 for 4-HNE stains), and then incubated with 3% H_2_O_2_ in methanol for 10 min to block endogenous peroxidase activity. Next, the unspecific binding sites were blocked with 2.5% normal horse serum (Vector Laboratories, Burlingame, CA, USA) for 30 min. The sections were incubated overnight at 4 °C with rabbit polyclonal anti-rat antibodies against TNFα (#AF7014, Affinity Biosciences, Cincinnati, OH, USA) diluted 1:200 in phosphate-buffered saline (PBS), or polyclonal antibodies against 4-HNE (#bs-6313R, Bioss, Woburn, MA, USA) diluted 1:400 in PBS. Sections were incubated with secondary antibodies (ImmPRESS Universal reagent Anti-Rabbit Ig, Vector Laboratories) for 30 min. The specificity of immunohistochemical staining was checked by omitting the primary antibody and by replacing it with the rabbit serum. The sections were visualized with Liquid DAB + Substrate Chromogen System (Dako, Carpinteria, CA, USA), then counterstained with hematoxylin (Sigma-Aldrich, Merck KGaA, St. Louis, MO, USA), dehydrated in ethanol series, rinsed in xylene, and mounted in DPX (Sigma-Aldrich). The labelled tissues were photographed using an SC-50 camera (Olympus Corp.) mounted on an Olympus Cell-Vivo IX 83 microscope (Olympus Corp.). Analyses were performed for 4–7 animals per group, in 6 non-overlapping fields (microphotographs) for each animal, using the cellSens Dimension 4.1 software (Olympus Corp.), as described for PAS-stained area analysis.

### 4.5. qPCR Array

A dedicated PCR array (Rat Stress & Toxicity Pathway Finder, #330231 PARN-003ZA, Qiagen, Hilden, Germany) was used to assess whether fenofibrate affects molecular pathways responsive to cellular stress insults. 2 μg total RNA, obtained with the Total RNA Mini kit (A&A Biotechnology, Gdynia, Poland), was reverse transcribed using the RevertAid Reverse Transcriptase (Thermo Fisher Scientific, Fitchburg, WI, USA) and oligo(dT)_18_ primers (Sigma-Aldrich, Merck KGaA, St. Louis, MO, USA). The PCR array was performed in the StepOnePlus Real-Time PCR System (Applied Biosystems, Carlsbad, CA, USA), following the manufacturer’s instructions, using spools of 5× diluted cDNA from 5 rats for each of the six rat groups studied. The gene expression levels were normalized to Hypoxanthine Phosphoribosyltransferase 1 (*Hprt1*) and Ribosomal Protein, Large, P1 (*Rplp1*), selected for their most stable expression from the set of manufacturer-defined housekeeping genes amplified in the same plate.

### 4.6. Quantification of Gene Expression Using Real-Time PCR

The real-time PCR technique was used to verify selected results of the PCR array, as well as to relatively quantify the mRNA expression of additional targets. RNA obtained from homogenized kidney samples with Total RNA Mini kit (A&A Biotechnology, Gdynia, Poland) was reverse transcribed with RevertAid Reverse Transcriptase (Thermo Fischer Scientific, Fitchburg, WI, USA). The PCR assays were performed using 5× diluted cDNA samples (7–8 per group), AmplifyMe SG Universal Mix (Blirt, Gdańsk, Poland), and 0.25 µM sense and antisense primers (Sigma-Aldrich, Merck KGaA, St. Louis, MO, USA). Gene expression was quantified relative to the geometric mean of *Hprt1* and *Rplp1* expression as internal control, using the 2^−ΔΔCT^ method. The primer sequences, designed using PrimerBLAST (NCBI, National Center for Biotechnology Information, Bethesda, MD, USA), are available in Appendix A.

### 4.7. Assessment of Protein Content by Western Blotting

Lysates (5–6 per group) were prepared as whole-cell lysates with Mammalian Cell Extraction Kit (BioVision, Miliptas, CA, USA) or nuclear and cytoplasmic extracts with NE-PER^®^ Nuclear and Cytoplasmic Extraction Kit (Thermo Fisher Scientific, Fitchburg, WI, USA), with the addition of protease and phosphatase inhibitors (#P8340, Sigma-Aldrich, Merck KGaA, St. Louis, MO, USA, and #78420, Thermo Scientific, respectively). 30 μg protein samples (or 12 μg for nuclear extracts) were separated on 10% SDS-PAGE gels, transferred to PVDF membranes (Bio-Rad, Warsaw, Poland), and then blocked with 7% non-fat milk in TBS with 0.1% Tween20 (TBST) for 1 h. The membranes were incubated overnight at 4 °C with primary antibodies (diluted in 3% non-fat milk in TBST) produced in rabbit: PhosphoPlus^®^ AMPKα (Thr172) antibody duet (#8208S; Cell Signaling Technology, Leiden, The Netherlands), IL-6 (#bs-4539R, Bioss), Nlrp3 (#bs-10021R, Bioss), Nrf2 (#bs-1074R, Bioss), Pgc1α (#20658-1-AP, Proteintech, Rosemont, IL, USA), Pparα (#AF5301, Affinity Biosciences), Sesn2 (#10795-1-AP, Proteintech), Sirt1 (#bs-0921R, Bioss), Vegfa (#CSB-PA08249A0Rb, CusaBio); or antibodies produced in mouse: Acadm (#67742-1-Ig-SS, Proteintech). After washing, the membranes were incubated for 1.5 h with horseradish peroxidase-conjugated secondary antibodies (anti-rabbit IgG-peroxidase, 1:10,000; #A6154, Sigma-Aldrich; anti-mouse IgG-peroxidase, 1:20,000; #A9044, Sigma-Aldrich). The levels of β-actin (peroxidase-conjugated IgM antibody, 1:50,000, 1 h at room temperature; #SC-47778, Santa Cruz Biotechnology, Inc., Santa Cruz, CA, USA) or GAPDH (peroxidase-conjugated antibody, 1:50,000, 1 h at room temperature; #AB2302, Merck Millipore) served as the loading control for whole-cell lysates and cytoplasmic fractions, whereas the lamin B1 antibody (#A1910, ABclonal, Woburn, MA, USA) was the loading control for nuclear fractions. Bands were visualized using Clarity Western ECL Substrate (Bio-Rad, Hercules, CA, USA) in ImageQuant™ LAS 500 chemiluminescence CCD camera (Cytiva Europe GmbH, Freiburg im Breisgau, Germany) and quantified by densitometric analysis using the ImageQuant™ TL 10.0 software (Cytiva Europe GmbH).

### 4.8. Statistical Analyses

Statistical analyses were performed using GraphPad Prism v.6 (GraphPad Software, San Diego, CA, USA). Because not all sets of data followed normal distribution (Shapiro–Wilk test), statistical analyses were performed using the nonparametric Kruskal–Wallis ANOVA with Dunn’s multiple comparisons test or Mann–Whitney U test, following Grubb’s test for outliers. Data are presented as mean ± standard error (SEM). Statistical significance was set at *p* < 0.05. Statistically significant differences are denoted with asterisks: * *p* < 0.05, ** *p* < 0.01, *** *p* < 0.001, or with hashtags to indicate differences between corresponding groups of young and old rats: # *p* < 0.05, ## *p* < 0.01, ### *p* < 0.001. Data obtained in the two separately conducted experiments testing the effects of 0.1% or 0.5% FF were combined, because there were no significant differences between the control groups in the two experiments.

## 5. Conclusions

Our results show divergent, age-related effects of fenofibrate on the structure and on molecular stress-associated targets in the rat kidney. In the young kidney, fenofibrate, especially at a low dose, may reduce interstitial fibrosis and activate antioxidant defenses, with variable modulation of inflammatory response effectors. In the aged kidney, low-dose fenofibrate may reduce the age-related thickening of basal laminae and activate fatty acid oxidation regulators and gene expression. However, in the kidney of old rats, the activation of antioxidant defenses is impaired, while some proinflammatory genes’ expression is increased upon high-dose fenofibrate treatment. These results obtained in the rat model imply that fenofibrate therapy in the elderly should be applied with special caution and monitoring of kidney function parameters.

## Figures and Tables

**Figure 1 ijms-25-03038-f001:**
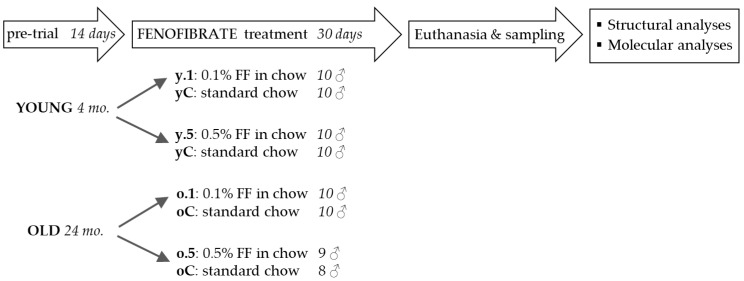
Experimental design in the study: In separate experiments, young and old rats were fed either 0.1% or 0.5% (*w*/*w*) fenofibrate, mixed into chow before pelleting, while control groups received the same chow without the drug. After 30 days, the animals were sacrificed and tissues were collected for subsequent structural and molecular analyses. The control groups for rats of either age were combined after checking that there were no significant differences between them. FF, fenofibrate; y.1, o.1—young or old rats treated with 0.1% fenofibrate; y.5, o.5—young or old rats treated with 0.5% fenofibrate; yC, oC—young or old control groups.

**Figure 2 ijms-25-03038-f002:**
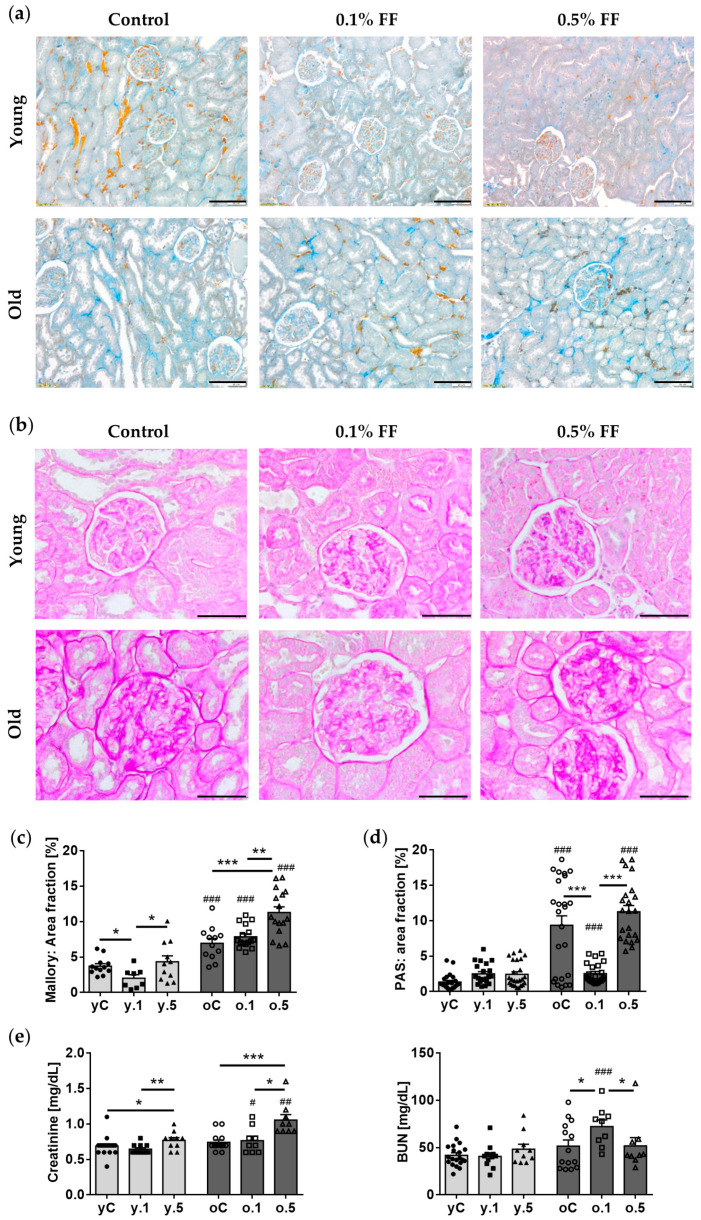
Effects of fenofibrate (FF) on the morphology and functional parameters of the kidney in young and old rats: (**a**) Mallory trichrome staining, scale bar 100 μm; (**b**) periodic acid-Schiff (PAS) staining, scale bar 100 μm; (**c**) quantification of collagenous tissue—Mallory-stained area fraction; (**d**) quantification of PAS-stained area fraction (expansion of intraglomerular and basement membrane staining); (**e**) serum creatinine and blood urea nitrogen (BUN) concentrations. Young (y) and old (o) rats for 30 days received 0.1% (y.1, o.1) or 0.5% FF (y.5, o.5) mixed into chow, or the same chow without supplementation in control groups (yC, oC). Circles, squares, and triangles correspond to values of individual rats, filled and open symbols refer to young or old rats, respectively. Data are means ± SEM of 4–7 (stains) or 6–13 (creatinine and BUN) animals per group: * *p* < 0.05, ** *p* < 0.01, *** *p* < 0.001 versus indicated subgroups in a given age group; hashtags indicate differences between corresponding groups of young and old rats: # *p* < 0.05, ## *p* < 0.01, ### *p* < 0.001.

**Figure 3 ijms-25-03038-f003:**
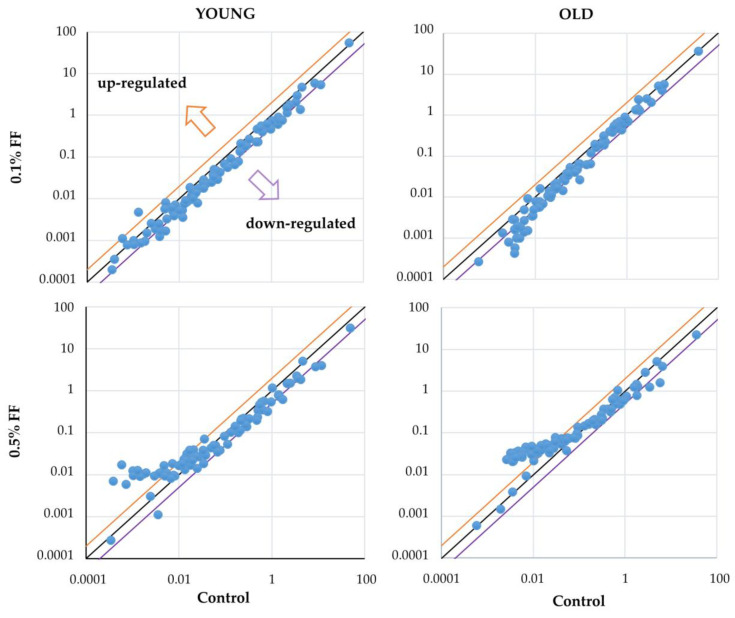
Two doses of fenofibrate (FF) differently affected the expression of stress-associated genes. A PCR array for 84 stress and toxicity-associated genes (Qiagen) was performed for each control and treatment group, using pooled mRNA from five animals per group. Scatter plots present the log10 of the expression level of each gene in treatment vs. control group. The orange and purple diagonal lines indicate the threshold of two-fold change.

**Figure 4 ijms-25-03038-f004:**
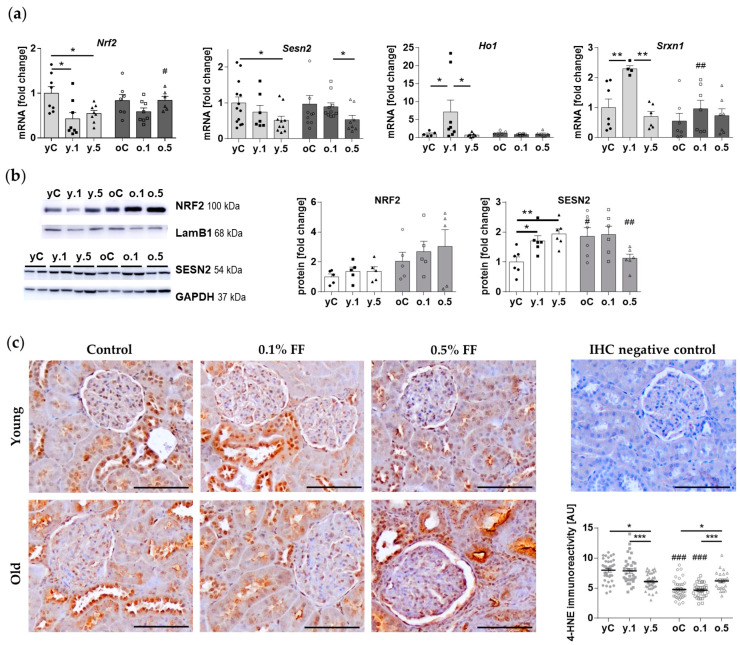
Fenofibrate differently affects the expression of oxidative stress-related factors in the young and old rats’ kidney: (**a**) mRNA expression was quantified using real-time PCR, relative to the mRNA levels of *Hprt1* and *Rplp1* as reference genes; (**b**) Representative Western blots and quantification of NRF2 (nuclear protein level, relative to lamin B1, LAMB1) and SESN2 (in whole-cell lysates, relative to GAPDH); (**c**) Immunohistochemical staining for 4-hydroxynonenal and quantification of the stained area (arbitrary units, AU). Values for individual animals are presented as described in the legend to Figure 2. Data in the graphs are means ± SEM of 5–6 (WB), 7–13 (qPCR), or 4–7 (IHC) animals per group: * *p* < 0.05, ** *p* < 0.01, *** *p* < 0.001 versus indicated subgroups in a given age group; hashtags indicate differences between corresponding groups of young and old rats: # *p* < 0.05, ## *p* < 0.01, ### *p* < 0.001.

**Figure 5 ijms-25-03038-f005:**
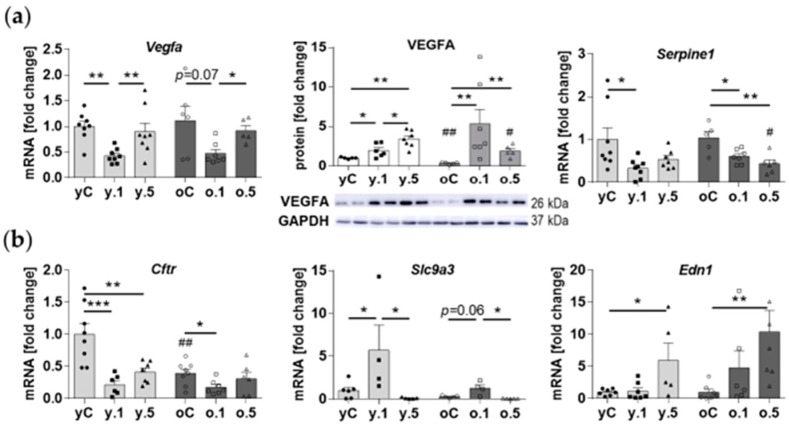
Fenofibrate modulates the expression of genes involved in hypoxia signaling and osmotic stress in the kidney of young and old rats: (**a**) Representative Western blots and quantification of VEGFA, and mRNA expression of *Vegfa* and *Serpine1*; (**b**) mRNA levels of genes encoding osmotic stress-related factors. Values for individual animals are presented as described in the legend to Figure 2. Data are means ± SEM of 5–6 (WB) or 6–8 (qPCR) animals per group; qPCR quantification relative to *Hprt1* and *Rplp1* as reference genes. * *p* < 0.05, ** *p* < 0.01, *** *p* < 0.001 versus indicated subgroups in a given age group; hashtags indicate differences between corresponding groups of young and old rats: # *p* < 0.05, ## *p* < 0.01.

**Figure 6 ijms-25-03038-f006:**
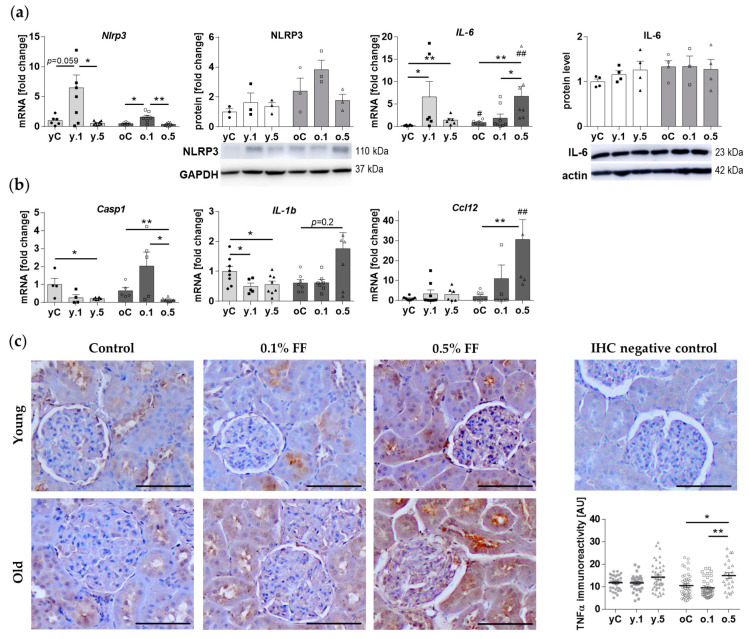
Fenofibrate modulates the expression of inflammatory genes in the young and old rats’ kidney: (**a**) mRNA and protein expression levels of NLRP3 and IL-6; representative Western blots are shown; (**b**) Gene expression of *Casp1*, *Il-1b* and *Ccl12*; (**c**) Immunohistochemical staining for TNFα and quantification of the stained area. Values for individual animals are presented as described in the legend to Figure 2. Data are means ± SEM of 3–4 (WB), 4–8 (qPCR) or 4–7 (IHC) animals per group: * *p* < 0.05, ** *p* < 0.01 versus indicated subgroups in a given age group; hashtags indicate differences between corresponding groups of young and old rats: # *p* < 0.05, ## *p* < 0.01. AU, arbitrary units.

**Figure 7 ijms-25-03038-f007:**
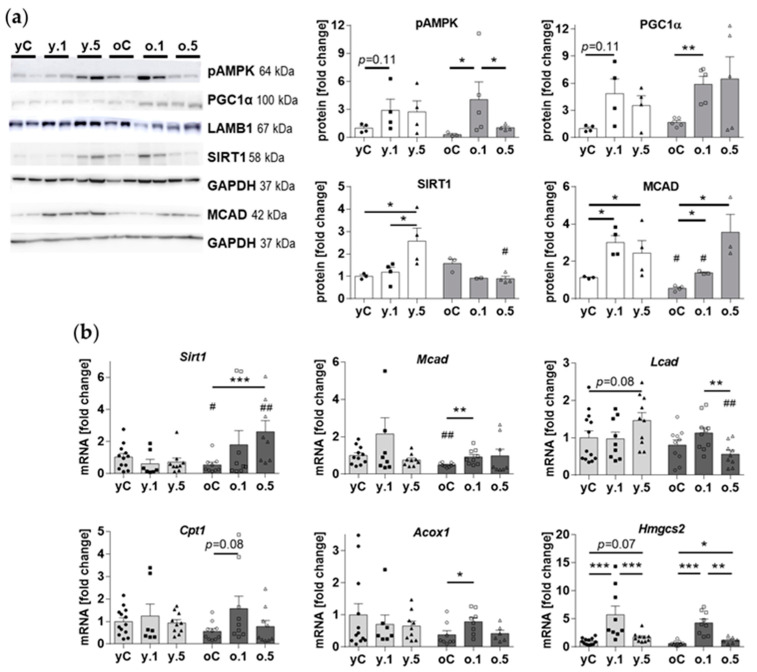
Low-dose fenofibrate increases the expression of oxidative metabolism-related enzymes in the kidney of young and old rats: (**a**) Representative Western blots and quantification of phosphorpho-AMPK, PGC-1α, SIRT1, and MCAD; (**b**) mRNA levels of genes encoding oxidative phosphorylation and ketogenic enzymes. Values for individual animals are presented as described in the legend to Figure 2. Data are means ± SEM of 3–5 (WB) or 7–8 (qPCR) animals per group: * *p* < 0.05, ** *p* < 0.01, *** *p* < 0.001 versus indicated subgroups in a given age group; hashtags indicate differences between corresponding groups of young and old rats: # *p* < 0.05, ## *p* < 0.01.

## Data Availability

The data that support the findings of this study are available from the corresponding author upon reasonable request.

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
