# Peer review of "High-Dose Fenofibrate Stimulates Multiple Cellular Stress Pathways in the Kidney of Old Rats"

_ijms, 2024, doi:10.3390/ijms25053038_

Round 1
Reviewer 1 Report
Comments and Suggestions for Authors
This is an interesting work, with an original approach, but it should be improved in some details.
Introduction: In lines 64 -68 the objective is mixed with a summary of the results. I think that at this point it would be more interesting to formulate a more complete objective and leave the results for the next section.
Results:
- Since in the format of this article the Materials and Methods section is at the end, it might be useful to include a small summary or diagram of the experimental groups so that the results can be easily understood without having to read the Materials and Methods section.
- Figure 1: the graphs appear cut off, and the quality of figure 6 is not very good and appears blurry
- GAPDH load control in figure 3 (western blot) does not seem like the best option to me since it alone shows differences, in fact the image is exactly the same as that of SESN2. Coincidence or typo? If for SESN2 it is considered that there are differences, for the loading control the same should be considered and therefore look for another one or repeat the WB since the same amounts of protein are not loaded in each well.
Discussion: This section is too long and confusing, the message of the manuscript is lost by seeking so many comparisons with the literature
Reviewer 2 Report
Comments and Suggestions for Authors
The study is potentially interesting as fenofibrate is a commonly used medication to treat hyperlipidaemias with elevated triglycredi levels in many diseases. Given the fact that age increases in the overall population, especially in those with chronic diseases, renal impairment is also more frequent in those who are treated with fenofibrate. Thus, the renal effects of chronic fenofibrate administration have a major clinical significance.
The manuscript is very well written, the experiments are well designed and the results are very interesting. Still, data presentation needs major improvements.
1) For instance, figures need major changes. In all figures, pls change bar charts to scatter plots showing individual data. In addition, half of the bar charts in Fig.1. were cut out, pls correct.
2) Fig 1a does not show any important information and could be omitted. Scale bar description in figures should be increased in size to be legible. Please improve the quality of Fig 1c, the photos look blurred and out of focus.
3) In Fig 3b, western blots lack the molecular weights, pls add to the blots.
4) In Fig 3c, the 4-HNE staining is mainly localized to renal tubular cells and their nuclei, but not to brush border as stated in the text. Please revise.
5) Gene names in all charts should be in italics like within the main text. Pls correct in all figures.
6) The TNFa IHC in Fig 5c does not seem specific, as stains mostly cell nuclei in all presented groups (nuclei of some tubules, podocytes, parietal epithelial cells). And even in the IHC negative control we can observe a weak intraglomerular unspecific staining. Moreover, the photos do not represent the difference seen in the bar chart between o1 and o5. Please revise this staining, its evaluation and the photographs! Maybe the problem is the TNFa antibody was diluted in PBS rather then in PBS-T containing blocking serum?
7) The quality of Figure 6 is overall bad and pixelated, please improve.
8) Was the amount of daily drug intake monitored at several time-points throughout the study (eg with pair-feeding)?
Round 2
Reviewer 1 Report
Comments and Suggestions for Authors
Unfortunately I can only consult the other reviewer's cover letter (in all the documents on the website I can only find the answers to reviewer 2) I cannot see the answers to my questions. I noticed that the authors have made some changes but I can't figure out why they still show a figure with a loading control that varies between samples
Author Response
Dear Reviewer,
Thank you for your time and effort to review our manuscript. We sincerely apologise for the lack of answers to your previous comments - apparently, they did not get uploaded properly. You will find these original answers attached to this message.
With regard to the fact that we use "a figure with a loading control that varies between samples", we deeply regret that we cannot amend this problem. Unfortunately, the blots for GAPDH after SESN2 measurement did show slight variation. For each blot, densitometric analyses were performed relative to GAPDH bands, and consistently showed differences between the young control and fenofibrate-treated groups. However, we are no longer able to repeat the blots to obtain more satisfactory results, as we have run out of the antibodies, the project budget has been spent, and - most importantly - the protein lysates after prolonged storage will show signs of degradation. We are deeply sorry for this shortcoming.
We do hope, however, that you will find the other answers to your original comments satisfactory.
On behalf of all the Authors,
Kind regards,
Agata Wrońska

Reviewer 2 Report
Comments and Suggestions for Authors
All my concerns have been addressed.
Author Response
Dear Reviewer,
Thank you for your time and effort in reviewing our manuscript, your valuable comments how to improve it, as well as the encouraging evaluation.
On behalf of all the authors,
Sincerely,
Agata Wrońska
Round 3
Reviewer 1 Report
Comments and Suggestions for Authors
The authors have made the required changes, or have explained the reason why some modifications cannot be made
Author Response
Dear Reviewer,
Thank you again for the time and effort you put into reviewing our manuscript. We very much appreciate your comments, which enabled us to improve our work.
Kind regards,
Agata Wrońska